# XSkill: Cross Embodiment Skill Discovery

**Mengda Xu** [1,2], **Zhenjia Xu** [1], **Cheng Chi** [1], **Manuela Veloso** [2,3], **Shuran Song** [1]

[1] Department of Computer Science, Columbia University

[2] J.P. Morgan AI Research [3] School of Computer Science, Carnegie Mellon University (emeritus)

**Abstract:** Human demonstration videos are a widely available data source for robot learning and an intuitive user interface for expressing desired behavior. However, directly extracting reusable robot manipulation skills from unstructured human videos is challenging due to the big embodiment difference and unobserved action parameters. To bridge this embodiment gap, this paper introduces XSkill, an imitation learning framework that 1) discovers a cross-embodiment representation called skill prototypes purely from unlabeled human and robot manipulation videos, 2) transfers the skill representation to robot actions using conditional diffusion policy, and finally, 3) composes the learned skill to accomplish unseen tasks specified by a human prompt video. Our experiments in simulation and real-world environments show that the discovered skill prototypes facilitate both skill transfer and composition for unseen tasks, resulting in a more general and scalable imitation learning framework. The benchmark, code, and qualitative results are on project website.

**Keywords:** Manipulation, Representation Learning, Cross-Embodiements

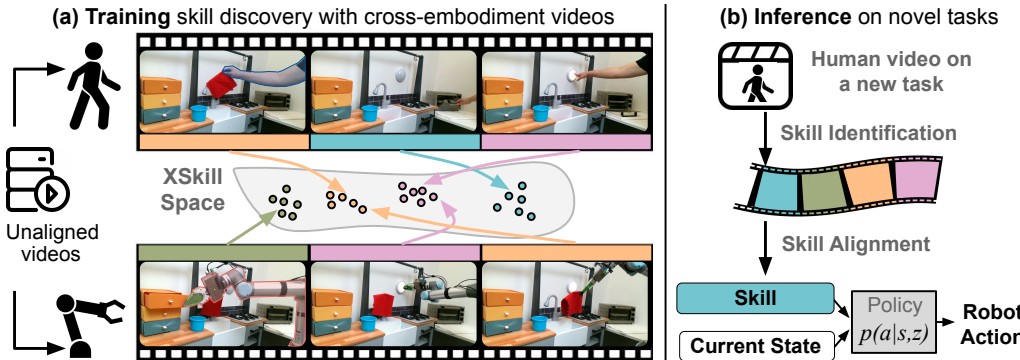

Figure 1: **Cross Embodiment Skill Discovery.** XSkill first learns a cross-embodiment skill representation space (XSkill Space on the left). During inference, given a human demonstration of unseen tasks, XSkill first identifies the human skills by projecting the video demonstration onto the learned cross-embodiment skill representation space. The identified skills are then executed by the skill-conditioned visuomotor policy.

## 1 Introduction

A successful imitation learning algorithm from human demonstration is enabled by three critical capabilities: 1) **Discover**, decomposing the demonstrated task into a set of sub-tasks and identifying the common and reusable skills required to accomplish these sub-tasks. 2) **Transfer**, mapping each of the observed skills to its own embodiment, which is different from that of the demonstrator. 3) **Compose**, performing novel compositions of the learned skills to accomplish new tasks.

This paper addresses these critical capabilities by decomposing and identifying appropriate "skills" from human demonstration so that they are *transferable* to robots and *composable* to perform new tasks. We refer to the task as **"Cross-Embodiment Skill Discovery"** and introduce our method

7th Conference on Robot Learning (CoRL 2023), Atlanta, USA.

**XSkill** for this task. At its core, XSkill learns a shared embedding space for the robot and human skills through self-supervised learning [1, 2, 3, 4, 5]. The algorithm extracts features for *unaligned* human and robot video sequences, where video clips share similar action effects (i.e., similar skill) should result in a closer feature distance. To encourage across-embodiment alignment, we introduce a set of learnable skill prototypes through feature clustering. These prototypes act as representative anchors in the continuous embedding space. By force to share the same set of prototypes, we could effectively align the skill representations between embodiments.

With the identified cross-embodiment skill prototypes, the robot can then learn a skill-conditioned visuomotor policy that transfers each identified skill to the robot's action space. During inference, the algorithm can one-shot generalize to new tasks using the learned skills, where the new task is defined by a single human demonstration (i.e., prompt video). With the proposed skill alignment transformer, the algorithm can robustly align skills in the human video to the robot visual observation, despite the embodiment difference and unexpected execution failures.

Our approach improves upon the direct imitation learning method [6] by decomposing the complex long-horizon tasks into a set of reusable skills (i.e., low-level visuomotor policies), which is much easier to learn and generalizable to new tasks through composition. Meanwhile, our approach differs from existing work on single-embodiment skill discovery [7, 8, 9], which solely relies on on-robot demonstration data. By learning cross-embodiment skill prototypes, our framework can use direct human demonstration, which is more cost-effective and scalable, even for non-expert demonstrators.

In summary, our contributions are as follows:
- We formulate the task of **cross-embodiment skill discovery**, a useful and essential building block for imitation learning. Together with the new cross-embodiment dataset in simulation and the real world, we hope to inspire future exploration in this area.
- Introducing the first attempt toward this task **XSkill** that consists of three novel components: 1) A self-supervised representation learning algorithm that discovers a set of share skill prototypes from the unlabeled robot and human videos. 2) A skill-conditioned diffusion policy that translates the observed human demonstration into robot actions. 3) A skill composition framework (with skill alignment transformer) to robustly detect, align and compose the learned skills to accomplish new tasks from a single human demonstration.

Our experiments in simulation and real-world environments show that the discovered skill prototypes facilitate both skill transfer and composition for unseen tasks, resulting in a more general and scalable imitation learning framework. The dataset and code will be publicly available.

## 2   Related Work

**Robot Skill Discovery.** A large body of works have been proposed for discovering robot skill via option frameworks [10, 7, 11, 12, 13, 14] or through the lens of mutual information [15, 8, 16, 17, 18, 19, 20]. Most of these works require interacting with the environment and yield high sample complexity. To ease the sample complexity, the other line of works [9, 21, 22, 23, 24] have explored discover skill directly from robot demonstration data. The majority of those prior works require physical state or robot action trajectories. BUDS [25] eases this requirement by discovering skills through raw RGB data. Unlike these prior works which discover skills only for a single embodiment, XSkill explores skill discovery in a cross-embodiment setting.

**Imitation learning.** Learning robot behavior from demonstration data is a longstanding challenge [6]. Prior works on imitation learning have shown promising results on real-world robot manipulation task through explicit policy [26, 27, 28, 29, 30, 31, 32, 33], implicit policy [34] or diffusion model [35, 36, 37]. Our works utilize the hierarchical imitation learning framework [38, 39, 40, 41, 42] to transfer the discovered skills through skill-conditioned diffusion policy[35]. XSkill can be categorized as one-shot imitation learning [43]. Most of the prior works imitate from the same-embodiment demonstration [44] or from a different embodiment [45, 46] but require task

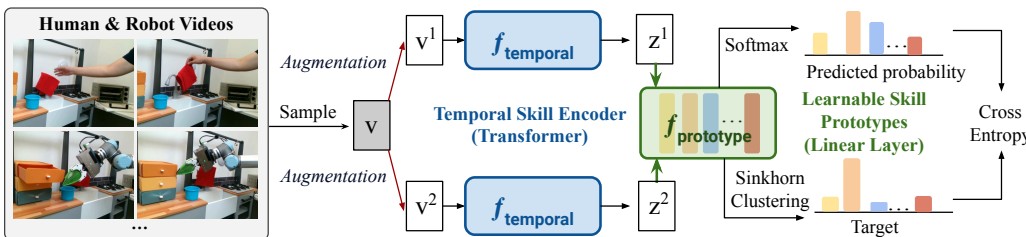

Figure 2: **XSkill Discover:** At each training iteration, a batch of video are sampled from the same embodiment dataset. Each video $v_i^t$ is augmented into two versions and encoded using temporal encoder $f_{\text{temporal}}$. The learnable skill prototypes $f_{\text{prototype}}$ are implemented as a normalized linear layer without bias. Both $f_{\text{temporal}}$ and $f_{\text{prototype}}$ are trained jointly to minimize the CorssEntorpy loss between the predicted and target the probability of skill prototypes. Sinkhorn regularization is applied to the target probability, ensuring all prototypes are used for each batch (same embodiment), thereby encouraging prototype sharing across embodiments.

label during the training. In contrast, XSkill does not require any task label or correspondence between embodiments during training to accomplish one-shot imitation from human demonstration.

**Learning from human video.** A number of works [47, 48, 49, 50, 51] have studied leveraging human videos to learn robotic policy. One approach is to construct reward functions from human video through domain translation [48, 52, 53, 54, 55, 56], video classifier [50, 57], or state representation learning [58, 49, 59]. Despite showing promising results in learning from cross-embodiment demonstration, most of these works require reinforcement learning (RL) to learn policy based on the constructed reward functions, which is expensive to deploy in the real world. In contrast to the majority of these works, our method does not involve RL in the loop and focuses on one-shot imitation from human videos. Bahl et al. [60] proposed to initialize policy through human prior but still requires interaction with the environment to improve the policy. Our work is related to Yu et al. [51, 61], which explored one-shot imitation through meta-learning. Unlike meta-learning approach, our method does not require any task pairing information during the training. More recently, MimicPlay [62] proposed leveraging human videos to learn a cross-embodiment plan latent space. While MimicPlay aims to minimize robot demo collection, our emphasis is on learning a cross-embodiment representation that allows us to reduce the reliance on robot demonstrations during inference.

## 3 Approach

The XSkill framework consists of three phases: Discover §3.1, Transfer §3.2, and Compose §3.3 that uses three different data sources. In the discover phase, the algorithm has access to a **human demonstration dataset** $\mathcal{D}^h$ and a **robot teleoperation dataset** $\mathcal{D}^r$ to discover a cross-embodiment skill representation space $\mathcal{Z}$. This space, along with $K$ common learnable skill prototypes, is learned through self-supervised learning. Both datasets are **unsegmented** and **unaligned** and each video in the dataset performs a subset of $N$ skills. In the transfer phase, the algorithm uses the robot teleoperation dataset $\mathcal{D}^r$ to learn the skill-conditioned visuomotor policy $P(a|s,z)$, where $z \in \mathcal{Z}$ and $s$ includes both robot proprioception and visual observation $o$. In the Compose phase, the algorithm takes as input a single **human prompt video** $\tau_{\text{prompt}}^h$ for a new task that requires an unseen composition of skills to complete. From this video prompt, the algorithm first identifies the order of skills used in the prompt and then composes the skills using the learned policy $P(a|s,z)$.

### 3.1 Discover: Learning Shared Skill Prototypes

As the first step, XSkill aims to discover skills in a self-supervised manner such that the learned visual representation of the **same skills** executed by **different embodiments** can be close in the cross-embodiment skill representation space $\mathcal{Z}$. Off-the-shelf vision representations are often insufficient since they are often sensitive to the visual appearance of the agent or environment. Instead, we want the learned skill representation to focus on the underlying skills being performed. To achieve this goal, XSkill introduces two key ideas:

- Learning a set of shared skill prototypes through soft-assignment clustering. These discrete prototypes act as representative anchors in the continuous embedding space. By forcing the use of shared prototypes, we can effectively align skill representations between embodiments.

- Regularizing the training process using Sinkhorn-Knopp clustering [63, 1] within single-embodiment batches. Together, they ensure all prototypes are used for each batch (all from the same embodiment). This design avoids the degeneration case where different embodiment maps to different prototypes, thereby ensuring prototype sharing.

**Skill representations.** XSkill extracts skill representations $z$ using **human and robot videos** from $\mathcal{D}^h$ and $\mathcal{D}^r$, mapping them into a shared representation space $\mathcal{Z}$. To mitigate variations in execution speed across different embodiments, we sample $M$ frames uniformly from each video $V_i$ and construct video clips $\{v_{ij}\}_{j=0}^M$ using a moving window of length $L$. Then, we extract the skill representation $z_{ij} = f_{\text{temporal}}(v_{ij})$ from each video clip with a temporal skill encoder consisting of a vision backbone and a transformer encoder [64]. We append a learnable representation token [65, 66] into the sequence to better capture the motion across frames.

**Skills as Prototypes.** Once the skill representations are obtained from demonstration videos, XSkill maps representations from all embodiments to a set of $K$ **skill prototypes** $\{c_k\}_{k=1}^K$, where each is a learnable vector. The skill prototypes are implemented as a normalized linear layer $f_{\text{prototype}}$ without bias. This mapping is accomplished through a self-supervised learning framework similar to SwAV [1]. The XSkill starts by augmenting the video clip $v_{ij}$ using a randomly selected transformation before feeding into $f_{\text{temporal}}$ (e.g., random crop). Subsequently, XSkill projects the normalized representation $\frac{z_{ij}}{||z_{ij}||_2}$ onto the set of learnable skill prototypes $\{c_k\}_{k=1}^K$. The probability $p_{ij}$ of skills being executed in the given video clip $v_{ij}$ is predicted by applying the Softmax function. The target distribution $q_{ij}$ is obtained from the other augmented version of the same video clip. The target probability, instead of applying the Softmax function over projection, is obtained by running online clustering Sinkhorn-Knopp algorithm, which we describe later in the paper. Both $f_{\text{temporal}}$ and $f_{\text{prototype}}$ are trained jointly to minimize the CorssEntropy loss between the predicted $p_{ij}$ and target $q_{ij}$ skill prototypes distributions: $\mathcal{L}_{\text{prototype}} = -\sum_{i=1}^B \sum_{j=0}^M \sum_{k=1}^K q_{ij}^{(k)} \log p_{ij}^{(k)}$, where $B$ is batch size.

**Learning Aligned Skill Representation.** To ensure that the skill representation focuses on underlying skills rather than embodiment and is aligned across embodiments, XSkill employs a combination of data sampling and entropy regularization during the clustering process in training. In each training iteration, XSkill samples video clips from the same embodiment and constructs a batch. This batch is then fed into the framework shown in Fig. 2, where clustering is performed on the features from the same embodiment, disregarding the embodiment differences. A significant challenge arises as the skill embedding space might be segmented by embodiment, with skill representations for each embodiment occupying distinct regions in the embedding space. To address this issue, our goal is to enable the skill representation for each embodiment to fully utilize the entire embedding space. By allowing different embodiments to share each region in the space, the clustering algorithm is compelled to group representations based on the effect of the skill, resulting in an aligned skill representation space. We approach this as an entropy-regularized clustering problem, which can be efficiently solved using the Sinkhorn clustering algorithm. Further details and pseudocode for the Sinkhorn-Knopp can be found in the appendix.

**Time Contrastive Learning.** XSkill utilizes a time contrastive loss [67, 68] in order to encapsulate the temporal effects of skills within video demonstrations. It posits that skill prototype probabilities should be similar for video clips closer in time and dissimilar for those farther apart. This is achieved by establishing a positive window $w_p$ and a negative window $w_n$. For a given clip $v_{ix}$ at time $x$ in video $V_i$, a positive sample $v_{iy}$ is chosen within $w_p$ from time $x$, and a negative sample $v_{iz}$ is selected outside $w_n$. XSkill minimizes the following InfoNCE loss [69]:
$\mathcal{L}_{tcn} = -\sum_{i=1}^B \log \frac{\exp(\mathcal{S}(p_{ix}, p_{iy})/\tau_{\text{tcn}})}{\exp(\mathcal{S}(p_{ix}, p_{iy})/\tau_{\text{tcn}}) + \exp(\mathcal{S}(p_{ix}, p_{iz})/\tau_{\text{tcn}})}$, wherein $\mathcal{S}$ is the measure of similarity, which is implemented as dot product in XSkill and $\tau_{\text{tcn}}$ is the temperature. Here, $p_{ix}$, $p_{iy}$, $p_{iz}$ represent the skill prototype probabilities for clips $v_{ix}$, $v_{iy}$, and $v_{iz}$, respectively.

### 3.2 Transfer: Skill conditioned imitation learning

To transfer skill representations into concrete robot actions, we train a skill-conditioned visuomotor policy using imitation learning. In theory, any imitation learning policy can be used with the XSkill

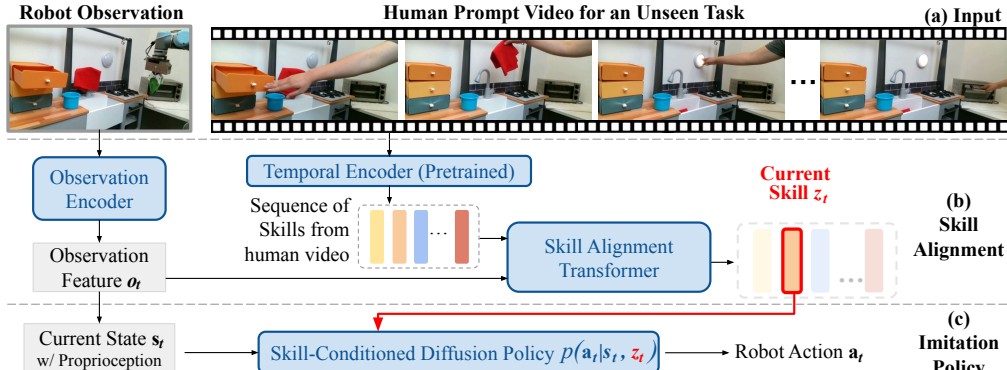

Figure 3: **Transfer & Composition:** During inference, a human demonstration of a new task is given, XSkill first extracts a sequence of skills, which can be viewed as a high-level task plan. However, this plan is not immediately aligned with robot execution speed due to the embodiment gap. Therefore we need to align the plan based on the robot's current observation, which is achieved by the Skill Alignment Transformer. The inferred skills are then passed into a skill-conditioned diffusion policy to get the robot's actions.

framework. In practice, we prefer to use diffusion policy as it achieves state-of-the-art results in many existing benchmarks. More specifically, our approach builds upon diffusion policy [35] which uses Denoising Diffusion Probabilistic Models (DDPMs) [70] to represent the multimodal action distributions from human teleoperation demonstration. Diffusion policy has been shown to be stable to train and work well with a small amount of data, both are essential for our tasks.

Our diffusion-based imitation learning policy $p(\mathbf{a_t}|s_t, z_t)$ is trained with robot teleoperation dataset $\mathcal{D}^r$, where $\mathbf{a_t}$ denotes an action-sequence $\{a_t, ..., a_{t+L}\}$ of length $L$ starting from state $s_t$. The diffusion policy takes skill representation $z_t$ and state $s_t$ which includes robot proprioception and visual observation $o_t$ as input and produces an action-sequence $\mathbf{a_t}$. The skill representation $z_t$ is computed with trained $f_{\text{temporal}}$ using $v_t = \{o_t, o_{t+1}..., o_{t+L}\}$ as we described in §3.1.

### 3.3 Compose: Performing unseen task from one-shot human prompt video

Once skills have been discovered and transferred into robot manipulations through imitation learning, our object is to compose the skills to solve unseen tasks based on a human prompt video which contains a demonstration by a human on how to complete an unseen task. To do so, XSkill first maps a human prompt video with length $T_{\text{prompt}}$ into the cross-embodiment skill representation space $\mathcal{Z}$ using learned $f_{\text{temporal}}$. This generates a sequence of skill representation for the demonstrated task, denoted as $\tilde{z} = \{z_t\}_{t=0}^{T_{\text{prompt}}}$, which is essentially a task execution plan. The robot can complete the task by sequentially executing the skills in the plan by querying the skill-conditioned diffusion policy.

However, directly following the skill sequence $\tilde{z}$ for execution often results in a fragile system that is sensitive to unexpected failures or speed mismatch. For example, if it fails to open a light, it needs to retry the skill to succeed. If the robot sequentially executes the $\tilde{z}$, it will proceed to execute the next skill without correcting the error. Therefore to improve the system's robustness, we introduce a Skill Alignment Transformer (SAT) that is described below.

**Skill Alignment Transformer (SAT).** The Skill Alignment Transformer, denoted as $\phi(z_t|o_t, \tilde{z})$, aligns the robot with the intended skill execution based on its current state. The key idea is that by analyzing the current state and comparing it to the skill sequence $\tilde{z}$, the robot can determine which part of the skills has already been executed and infer the most likely skill to be executed next. The cluster structure created by discrete prototypes aids in facilitating skill identification by SAT during inference time. This alignment process allows the robot to synchronize its task execution with the demonstrated task in the human prompt, thereby minimizing discrepancies arising from variations in execution speed between robots and humans and offering robustness against execution failures.

As illustrated in Fig. 3, SAT regards each skill in the skill sequence $\tilde{z}$ as a skill token. It employs a state encoder, $f_{\text{state-encoder}}$, to convert the robot's current visual observation into a state token. The skill and state tokens with position encoding, are then passed into a transformer encoder to predict the skill that needs to be executed next. Within the transformer, the state token can attend to each

skill token to determine whether the skill has been executed given the current state information. For example, if the light is already open in the current state, the skill to open the light should not be considered the next skill to be executed.

To train SAT $\phi(z_t|o_t, \tilde{z})$, we sample a full trajectory from the robot teleoperation dataset and extract the skill sequence $\tilde{z}$. This is achieved by passing robot trajectory video clips $\{v_t\}_{t=0}^{T}$, where $v_t = \{o_t, o_{t+1}..., o_{t+L}\}$, through the temporal skill encoder, yielding $\{z_t\}_{t=0}^{T}$. The length of the sampled trajectory is represented by $T$. Next, a time index $t$ is chosen randomly within the range $[0, T]$. Our system predicts $\hat{z}_t$ using the skill sequence $\tilde{z}$ and visual observation $o_t$ as inputs to SAT. We optimize the model by minimizing the mean square error between $\hat{z}_t$ and the actual $z_t$.

## 4 Evaluation

**Environment.** We test XSkill on both simulated and real-world environments:

- **Franka Kitchen:** is a simulated kitchen environment [71] that includes 7 sub-tasks and is accompanied by 580 robot demonstration trajectories. To create a cross-embodiment demonstration, we construct a sphere agent that is visually significantly different from the original robot. To further increase the domain gap, we sub-sample sphere agent demonstrations to emulate the execution speed differences. During the inference, the robot must complete an unseen composition of sub-tasks after viewing a prompt video from the sphere agent demonstration.
- **Realworld Kitchen:** is a new benchmark we introduce to evaluate algorithm performance on physical robot hardware. The dataset contains four sub-tasks, namely, opening the oven, grasping a cloth, closing a drawer, and turning on a light. We have recorded 175 human demonstrations and 175 teleoperation demonstrations. Each demonstration completes three sub-tasks in a randomly determined order. During inference, the robot is required to complete an unseen composition of either three or four sub-tasks after observing a prompt video taken from a human demonstration.

**Baselines.** We compare XSkill with the following baselines:

- **GCD Policy:** Instead of using skill-conditioned policy, we compare to a goal-conditioned diffusion policy $\pi(\mathbf{a_t}|s_t, g_t)$, where the goal image $g_t$ is the image in prompt video $H$ steps after the current time $t$ after alignment. The alignment is done using nearest-neighbor matching between robot observation and prompt video in the embedding space, where the embedding is trained jointly with the policy.
- **GCD Policy w. TCN:** Same as the GCD Policy above but replacing the video encoder with pre-trained Time-Contrastive Network (TCN)[67].
- **XSkill w. NN-composition:** XSkill removing skill alignment transformer. Instead, find the alignment using the nearest neighbor image between the robot observation and the prompt video. The image embedding is extracted using the same encoder $f_{\text{temporal}}$ as XSkill.
- **XSkill w.o proto. loss:** XSkill removing prototype loss $\mathcal{L}_{\text{prototype}}$ in the discover phase.

**Implementation Details.** We set the number of prototypes $K$ as 128 in the simulated environment and 32 prototypes in the real-world environment. The video clip length $L$ and uniform sample frames $M$ are set as 8 and 100 for both simulated and real-world kitchens. The ablation study on $K$, time contrastive loss, and more implementation details can be found in the supplementary material.

**Evaluation protocol.** During inference, the robot is required to accomplish the sub-tasks in the *same order* as demonstrated in the prompt video. The performance of XSkill and all baseline methods is evaluated based on both sub-task completion and order of completion. If the robot executes an undemonstrated sub-task, the episode ends. The evaluation metric is denoted

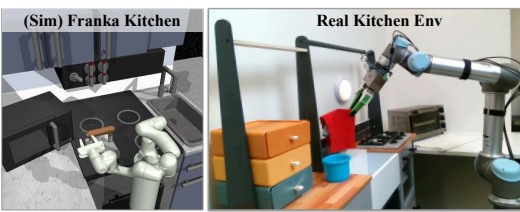

Figure 4: **Evaluation Environments.**

as $\frac{\text{number of sub-tasks completed}}{\text{number of total sub-tasks}}$. In the simulation, each method is trained using three distinct seeds and tested under 32 unique initial environment conditions during the inference. In the real-world kitchen, each task is assessed 10 times under varying initial environment conditions during the inference.

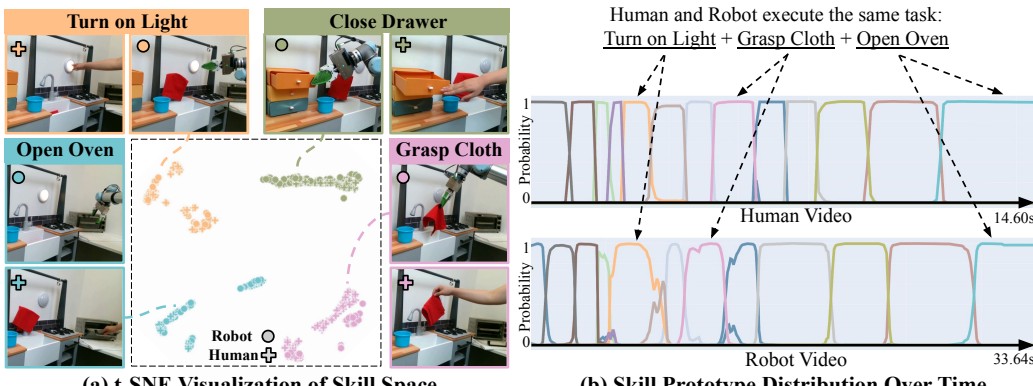

**(a) t-SNE Visualization of Skill Space**       **(b) Skill Prototype Distribution Over Time**

Figure 5: **XSkill embedding.** **(a)** We utilize t-SNE visualization to showcase the alignment of skill representations among various embodiments when in contact with the same object. **(b)** We present projected prototypes for both humans and robots executing identical tasks. XSkill achieves efficient alignment of representations, not just during physical contact, but also during the transition between manipulating different objects.

## 4.1 Key Findings

**XSkill can learn cross-embodiment skill representation.** [XSkill] learns cross-embodiment skill representation by effectively extracting reusable skills from demonstrations including both in-contact manipulation with various objects and also the transitions between them. In Fig. 5(a), we visualize these learned skill representations using t-SNE, where the skills executed by the human and robot to manipulate the same object are grouped together and clearly separated from the others. Additionally, we visualize the projected prototypes for human and robot completion of the same task in Fig. 5(b). Despite differences in execution speed (humans execute $\times$ 2 faster), [XSkill] can decompose the task into meaningful and aligned skill prototypes for both in-contact manipulations and transitions between them. Consequently, the performance of [XSkill] with cross-embodiment prompts only drops around $5\%$, compared to using the same embodiment prompt (Tab. 2 seen tasks).

**XSkill can generalize the imitation policy to unseen tasks.** [XSkill] achieves 70.2% and 60% success (Tab. 1 & 2) on unseen tasks with cross-embodiment prompts in simulated and real-world environments, which outperforms all baselines. As shown in Fig. 6 (a), [XSkill] is capable of decomposing unseen tasks

Table 1: **Simulation Result (%)**

|  | Same | Cross Embodiment | | | Avg |
| --- | --- | --- | --- | --- | --- |
| Execution speed | $\times$ 1 | $\times$ 1 | $\times$ 1.3 | $\times$ 1.5 | / |
| GCD Policy | 91.4 | 0.00 | 0.00 | 0.00 | 22.8 |
| GCD Policy w. TCN | 2.50 | 3.55 | 2.00 | 1.25 | 2.32 |
| XSkill w. NN-compose | 93.7 | 61.2 | 23.4 | 15.2 | 48.4 |
| XSkill w.o proto. loss | 80.1 | 56.3 | 12.5 | 3.75 | 38.2 |
| XSkill | **95.8** | **89.4** | **83.7** | **70.2** | **84.8** |

into sequences of previously seen skill abstractions which can be executable by the learned skill-conditioned imitation policy. As a result, [XSkill] enables one-shot imitation learning through skill decomposition and re-composition. The performance of [XSkill] slightly drops in the real world due to novel transition dynamics presented in prompts and a scarcity of collected robot data. For instance, the robot struggles to complete tasks involving grasping the cloth followed by closing the drawer, since no such transition dynamics are present in the collected robot teleoperation dataset. In summary, [XSkill] demonstrates promising task generalization, but its efficacy is still limited by the diversity in the robot teleoperation data.

**Skill prototypes are essential for cross-embodiment learning.** To assess the importance of shared skill prototypes, we compare our approach with [XSkill w.o proto. loss] in simulation. [XSkill] outperforms the baseline significantly with cross-embodiment prompts. Further, we observe that the performance of [XSkill w.o proto. loss]

Table 2: **Real-world Result (%)**

|  | 3 Subtasks | | | | 4 Subtasks | Avg |
| --- | --- | --- | --- | --- | --- | --- |
|  | Same | | Cross | | Cross | / |
|  | Seen | Unseen | Seen | Unseen | Unseen | / |
| GCD policy | 68.3 | 53.3 | 0.00 | 0.00 | 0.00 | 24.3 |
| GCD w. TCN | 25.0 | 22.2 | 26.7 | 23.3 | 15.6 | 22.6 |
| XSkill | **86.7** | **80.0** | **81.7** | **76.7** | **60.0** | **77.0** |

deteriorates rapidly when the cross-embodiment agent operates at a higher speed. These results suggest that skill prototypes not only facilitate the learning of morphology-invariant representations but also avoid learning a speed-sensitive one.

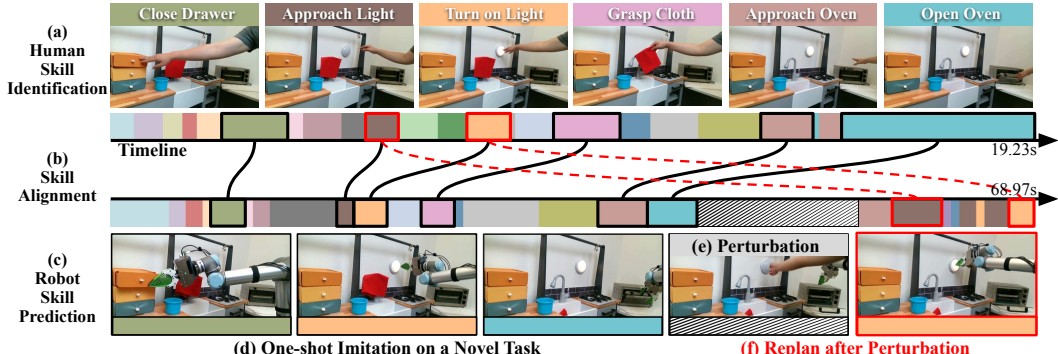

Figure 6: **Execution on a novel task and robustness to perturbation.** **(a)** XSkill analyzes a human video of a novel task, identifying skills for each timestep (represented by distinct colors). **(c)** The robot leverages this analysis to predict the appropriate skill based on the current observation and subsequently executes the corresponding skill. Skill alignment **(b)** is critical to handle execution speed difference caused by cross embodiment. **(e)** With appropriate skill conditions at each step, the robot achieves one-shot imitation of the novel task. **(e)** Deliberately introducing a perturbation, a human manually turns off the light. **(f)** The robot accurately predicts the necessary skills and adaptively replans the execution to successfully reach the goal state once again. Please check out the project website for videos

**SAT can align skills based on task progress.** [XSkill] outperforms [XSkill w. NN-composition] by more than $50\%$ with cross-embodiment prompt. The performance of NN-composition significantly declines when the cross-embodiment agent executes at a much faster speed (Tab. 1). We down-sample cross-embodiment demonstrations and also prompt videos with different ratio. For instance, a ratio of $\times 1.5$ emulates that human execute $\times 1.5$ faster than the robot in the real world. This can be attributed to two main factors. Firstly, relying solely on visual representation is prone to distraction due to morphological differences. Second, when the demonstration speed in the prompt videos is much faster than the robot's execution speed, certain states might not be captured in the prompt video. As a result, the retrieved skills can become inaccurate and unstable. In contrast, as illustrated in Fig. 6, the SAT enhances the robustness of [XSkill] to the demonstration speed and enables adaptive adjustment of skills based on the current state of the robot and the task progress.

**Sequential input benefits cross-embodiment learning.** Unlike [GCD Policy w. TCN], which relies on TCN encoding for single images, [XSkill] utilizes sequential image input to generate representations. In the cross-embodiment scenario, [GCD Policy w. TCN] outperforms vanilla [GCD Policy], indicating partial bridging of the embodiment gap through TCN embedding. However, [GCD Policy w. TCN] cannot follow the sub-tasks completion order in the prompt video, resulting in a lower evaluation score. This suggests that TCN embedding does not completely align the representations.

## 4.2 Limitation and Future Work

One limitation of XSkill is the requirement of specifying the number of prototypes as an input to the algorithm. While our ablation study demonstrates that XSkill is not highly sensitive to this number, fine-tuning hyperparameters may be necessary for optimal performance based on the dataset and its use. In addition, XSkill doesn't demand labeled correspondence between human and robot datasets. However, our current benchmark only comprises videos from the same laboratory camera setup. Future research could investigate the applicability of our approach in more diverse camera setups and environments, leveraging readily available YouTube videos and multi-environment datasets [72].

## 5 Conclusion

We introduce XSkill for the task of cross-embodiment skill discovery. This framework extracts common manipulation skills from unstructured human, robot videos in a way that is transferrable to robots and composable to perform new tasks. Extensive experiments in simulation and real-world demonstrate that XSkill improves upon the direct behavior cloning method especially complex long-horizon tasks. Moreover, by leveraging cross-embodiment skill prototypes, XSkill can directly leverage non-expert human demonstration for new tasks definition, making the framework much more cost-effective and scalable.

**Acknowledgments**

We would like to thank Zeyi Liu, Huy Ha, Mandi Zhao, Samir Yitzhak Gadre and Dominik Bauer for their helpful feedback and fruitful discussions.

Mengda Xu's work is supported by JPMorgan Chase & Co. This paper was prepared for information purposes in part by the Artificial Intelligence Research group of JPMorgan Chase & Co and its affiliates ("JP Morgan"), and is not a product of the Research Department of JP Morgan. JP Morgan makes no representation and warranty whatsoever and disclaims all liability, for the completeness, accuracy or reliability of the information contained herein. This document is not intended as investment research or investment advice, or a recommendation, offer or solicitation for the purchase or sale of any security, financial instrument, financial product or service, or to be used in any way for evaluating the merits of participating in any transaction, and shall not constitute a solicitation under any jurisdiction or to any person, if such solicitation under such jurisdiction or to such person would be unlawful. This work was supported in part by NSF Award #2143601, #2037101, and #2132519. We would like to thank Google for the UR5 robot hardware. The views and conclusions contained herein are those of the authors and should not be interpreted as necessarily representing the official policies, either expressed or implied, of the sponsors.

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

## Appendix

### A.1   Generalization to unseen transition

To evaluate the generalization capacity of XSkill, we conducted experiments within the simulation environment to assess its performance with unseen transitions. We designed two levels of task difficulty by excluding transitions used in the inference task from the training dataset. In Level 1, 25% of the transitions were removed, and in Level 2, 50% were removed. Notably, the main paper's experiment in the simulation was already conducted in the scenario involving the removal of 25% of transitions. The complete experiment results are summarized in Table A1.

Table A1: **Generalization Study Result (%)**

| | Level 1 | | | | Level 2 | | | |
|---|---|---|---|---|---|---|---|---|
| | Same | Cross Embodiment | | Overall | Same | Cross Embodiment | | Overall |
| Execution speed | $\times 1$ | $\times 1$ | $\times 1.5$ | / | $\times 1$ | $\times 1$ | $\times 1.5$ | / |
| XSkill K=128 | 95.8 | 89.4 | 70.2 | 85.1 | 62.5 | 55.0 | 47.5 | 55.0 |
| XSkill K=256 | 98.3 | 86.6 | 76.7 | 87.2 | 91.3 | 81.3 | 61.3 | 77.9 |
| XSkill K=512 | 97.5 | 90.7 | 71.8 | 86.7 | 87.5 | 84.4 | 67.5 | 79.8 |

The results of these experiments demonstrate XSkill's capacity to generalize to new tasks involving previously unseen skill compositions and transitions. XSkill achieved success rates of 90.7%

and $84.4\%$ in Level 1 and Level 2 tasks, respectively, when using the same-speed cross-embodiment prompt, and $71.8\%$ and $67.5\%$ for the cross-embodiment prompt at $\times 1.5$ speed. Further, we observe that increasing the number of prototypes $K$ enhanced the generalization potential. While various $K$ choices demonstrated comparable performance in Level 1 tasks, larger values of $K$ $(256, 512)$ significantly outperformed smaller ones in the more challenging Level 2 tasks, which lacked $50\%$ of the inference task transitions from the training data. We hypothesize that augmenting $K$ enhances the granularity of the representation space, thereby facilitating better generalization through interpolation.

## A.2 Ablation Study

### A.2.1 Number of skill prototypes

We performed an ablation study to assess the impact of the number of skill prototypes $(K)$ in our XSkill framework within the simulated Franka Kitchen environment. We tested $K$ values of 32, 128, 256, and 512, with the results for $K = 128$ reported in our main paper. The outcome of this ablation study can be found in Tab. A2.

We observed that increasing the number of skill prototypes $(K)$ to 256 or 512 did not degrade the performance of XSkill. However, reducing $K$ did impact performance adversely. We hypothesize that a smaller $K$ value (i.e., 32) may limit the representation capacity of the skill space, as all skill representations $z$ are enforced to map around one of the skill prototypes. This could potentially force distinct skills to map around the same prototype, resulting in diminished manipulation performance. On the contrary, a larger $K$ value doesn't hinder performance; in fact, increasing $K$ might augments the granularity of the representation space, allowing unique skills to have distinct representations within this space.

These results suggest that the performance of our framework is not significantly affected by the choice of $K$. We believe this is due to the fact that the projected skill prototypes are not directly inputted into the imitation learning policy $P(\mathbf{a_t}|s_t, z_t)$ and SAT $\phi(z_t|\tilde{z}, o_t)$. Instead, we utilize the continuous skill representation $z$ prior to projection. This choice allows for greater flexibility and granularity in representing skills, making the specific choice of $K$ less crucial. Therefore, while fine-tuning $K$ may still be necessary for optimal results in certain environments (e.g., a simulated Franka Kitchen with seven sub-tasks requiring large $K$ as opposed to a real-world kitchen with four sub-tasks where a $K$ value of 32 may suffice), our framework demonstrates robustness against variations in the number of skill prototypes.

Table A2: **Ablation: Number of $K$ (%)**

| Execution speed | Same $\times 1$ | Cross Embodiment $\times 1$ | Cross Embodiment $\times 1.5$ |
|---|---|---|---|
| XSkill $K = 32$ | 91.6 | 67.5 | 48.7 |
| XSkill $K = 128$ | 95.8 | 89.4 | 70.2 |
| XSkill $K = 256$ | 98.3 | 86.6 | 76.7 |
| XSkill $K = 512$ | 97.5 | 90.7 | 71.8 |

### A.2.2 Time contrastive loss

In order to demonstrate the significance of time contrastive loss, we conduct a comparative study with [XSkill w.o TC loss] using simulations. The results, presented in Tab. A3, clearly show a significant drop in performance for [XSkill w.o TC loss] compared to [XSkill], even under the same-embodiment setting. This empirical evidence underscores the vital role of time contrastive loss in enabling our representation to effectively capture the temporal effects of skills. Consequently, the learned skill representation with time contrastive loss is beneficial for downstream manipulation tasks.

Table A3: **Ablation: Time contrastive loss (%)**

| | Same | Cross Embodiment | | | Avg |
|---|---|---|---|---|---|
| Execution speed | $\times 1$ | $\times 1$ | $\times 1.3$ | $\times 1.5$ | / |
| XSkill w.o TC loss | 3.75 | 2.25 | 2.25 | 1.25 | 2.38 |
| XSkill | **95.8** | **89.4** | **83.7** | **70.2** | **84.8** |

## A.3 Additional Experiment Results

We present the performance results of XSkill for each task using cross-embodiment prompts during inference for the real-world environment in Table A4. When combined with the additional study on generalization in Appendix A1, a primary limitation for achieving generalization in real-world environments becomes apparent: the diversity present in robot teleoperation data.

Our observations indicate that XSkill is capable of generalizing to previously unseen transitions in simulations due to the sufficient and diverse nature of the collected data. This allows for effective interpolation and generalization. However, it's important to note that the data collected for the sub-task *Drawer* in the real-world environment lacks multi-modal properties, as illustrated in Figure 5 in the main paper. As a consequence, XSkill struggles to extend its capabilities to solve unseen transitions involving the *Drawer* sub-task.

Table A4: **XSkill Inference Task Per Task Results (%)**

| Inference Task | Cross Embodiment |
|---|---|
| Oven, Draw, Cloth | 80.0 |
| Draw, Cloth, Oven | 73.3 |
| Oven, Light, Cloth, Draw | 75.0 |
| Draw, Cloth, Light, Oven | 90.0 |
| Draw, Oven, Cloth, Light | 25.0 |
| Draw, Light, Cloth, Oven | 50.0 |

## A.4 Implementation Details

We have presented a summary of the three phases, namely Discover, Transfer, and Compose, in pseudocode. The pseudocode for each phase is provided in Algorithm 1, 2, and 3, respectively.

---

**Algorithm 1** Cross-embodiment Skill Discovery

---

1: **Input:** $K$: Number of skill prototypes. $\mathcal{D}^h$: Human demonstration dataset. $\mathcal{D}^r$: Robot teleoperation dataset
2: **Require:** $\mathcal{T}$: Random augmentation operation. *mm*: Matrix multiplication
3: **Require:** $f_{temporal}$: Temporal skill encoder. $C = [c_1, \ldots, c_K]$: $K$ Skill prototypes
4: **while** not converge **do**
5:      Sample a batch of video clips $v$ from $\mathcal{D}^h$ or $\mathcal{D}^r$
6:      $v_A = \mathcal{T}(v)$ and $v_B = \mathcal{T}(v)$          ▷ Compute two augmentations of $v$:
7:      $z_A = f_{\text{temporal}}(v_A)$ and $z_B = f_{\text{temporal}}(v_B)$      ▷ Compute skill representations
8:      $s_A = mm(z_A, C)$ and $s_B = mm(z_B, C)$      ▷ Compute projection
9:      $p_A = Softmax(s_A)$ and $p_B = Softmax(s_B)$      ▷ Predict skill prototypes probability
10:     $q_A = Sinkhorn(s_B)$ and $q_B = Sinkhorn(s_A)$      ▷ Compute target probability
11:     $\mathcal{L}_{proto} = \frac{1}{2}(CrossEntropy(p_A, q_A) + CrossEntropy(p_B, q_B))$      ▷ Compute prototype loss
12:     Sample positive and negative video clips for $v_A$: $v_A^{pos}, v_A^{neg}$      ▷ Or for $v_b$
13:     Compute associated skill prototypes probability $p_A^{pos}, p_A^{neg}$      ▷ Follow line 7 to line 9
14:     $\mathcal{L}_{tcn} = InfoNce(p_A, p_A^{pos}, p_A^{neg})$      ▷ Compute time contrastive loss
15:     $\mathcal{L}_{discovery} = \mathcal{L}_{proto} + \mathcal{L}_{tcn}$      ▷ Compute skill discovery loss
16:     Update $f_{\text{temporal}}$ and $C$      ▷ Update models
17: **end while**

---

---

**Algorithm 2** Cross-embodiment Skill Transfer

---

1: **Input:** $\mathcal{D}^r$: Robot teleoperation dataset.
2: **Require:** $f_{temporal}$: Learned temporal skill encoder (freeze).
3: **Require:** $\phi$: Skill Alignment Transformer (SAT). $p$: Imitation learning policy
4: **while** not converge **do**
5:     $\mathbf{o}, \mathbf{a}, \mathbf{s^{prop}} \sim \mathcal{D}^r$                      ▷ Sample a robot trajectory with length $T$
6:     $\tilde{z} = \text{Skill\_Identification}(\mathbf{o})$     ▷ Identify skills in video and form skill execution plan
7:     $t \sim [0, T]$                                ▷ Sample a time index
8:     $\hat{z}_t = \phi(\mathbf{o_t}, \tilde{z})$                   ▷ Predict the skill need to be executed at time $t$
9:     $\mathcal{L}_{SAT} = \text{MSE}(\hat{z}_t, \tilde{z}_t)$                          ▷ Compute SAT loss
10:    $\hat{\mathbf{a}_t} = p(\mathbf{o_t}, \mathbf{s_t^{prop}}, \tilde{z}_t)$        ▷ Predict the actions based on identified skill $\tilde{z}_t$ at time $t$
11:    $\mathcal{L}_{bc} = \text{MSE}(\hat{\mathbf{a}_t}, \mathbf{a}_t)$                      ▷ Compute behavior cloning loss
12:    $\mathcal{L}_{transfer} = \mathcal{L}_{SAT} + \mathcal{L}_{bc}$
13:    Update $\phi$ and $p$                        ▷ Update models based on transfer loss
14: **end while**

---

---

**Algorithm 3** Cross-embodiment Skill Compose (Inference)

---

1: **Input:** $\tau_{prompt}^h$: Human prompt video
2: **Require:** $\phi$: Learned Skill Alignment Transformer (SAT). $p$: Learned Imitation learning policy
3: $\tilde{z} = \text{Skill\_Identification}(\tau_{prompt}^h)$       ▷ Identify skills in video and form skill execution plan
4: **while** not success *or* episode not end **do**
5:     $\mathbf{o}_t, \mathbf{s}_t^{prop} = \text{env.get\_obs}()$     ▷ Get observation and robot proprioception from environment
6:     $\hat{z}_t = \phi(\mathbf{o_t}, \tilde{z})$                        ▷ Predict the skill need to be executed at time $t$
7:     $\hat{\mathbf{a}_t} = p(\mathbf{o_t}, \mathbf{s_t^{prop}}, \tilde{z}_t)$             ▷ Predict the actions based on predicted skill
8:     Execute the $\hat{\mathbf{a}_t}$ in environment.                      ▷ execute actions
9: **end while**

---

### A.4.1 Sinkhorn-Knopp Algorithm

XSkill employs the Sinkhorn algorithm to solve an entropy-regularized soft-assignment clustering procedure. As outlined in our main paper, our objective is to enhance cross-embodiment skill representation learning. We strive to ensure that each embodiment fully leverages the entire embedding space. Given that the skill prototypes serve as the cluster centroids, they essentially act as anchors in the representation space. We can efficiently realize this aim by uniformly soft-assigning all samples to every prototype.

Our goal is to project a batch of skill representation $\mathbf{Z} = [z_1, \ldots, z_B]$ onto the skill prototypes matrix $\mathbf{C} = [c_1, \ldots, c_K]$, where the columns of the matrix are $c_1, \ldots, c_K$. The intended code $\mathbf{Q} = [q_1, \ldots, q_B]$ or the target skill prototype probability should retain similarity with the projection while maintaining a specific entropy level. This can be expressed as an optimal transport problem with entropy regularization [63, 1]:

$$\max_{\mathbf{Q} \in \mathcal{Q}} \text{TR}\left(\mathbf{Q}^\top \mathbf{C}^\top \mathbf{Z}\right) + \varepsilon H(\mathbf{Q}) \tag{1}$$

The solution [63, 1] is given by:

$$\mathbf{Q}^* = \text{Diag}(\mathbf{u}) \exp\left(\frac{\mathbf{C}^\top \mathbf{Z}}{\varepsilon}\right) \text{Diag}(\mathbf{v}), \tag{2}$$

where $\mathbf{u}$ and $\mathbf{v}$ are normalization vectors, which can be iteratively computed using the Sinkhorn-Knopp algorithm. Sinkhorn-Knopp receives projection $\mathbf{C}^\top \mathbf{Z}$ as the input and iteratively modifies the matrix to satisfy the entropy regularization by producing double stochastic matrix. Through our experiments, we've observed that three iterations are sufficient. The Pytorch-like pseudocode for the Sinkhorn-Knopp can be found in pseudocode listing 1. The target skill probability for $z_i$ can be obtained from the $i^{th}$ column from the output $\mathbf{Q}^*$.

### A.4.2 Temporal Skill Encoder & Prototypes Layer

The temporal skill encoder $f_{\text{temporal}}$ consists of a vision backbone and a transformer encoder. To efficiently process a large batch of images, we employ a straightforward 3-layer CNN network followed by an MLP layer as our vision backbone. This network can be trained on a single NVIDIA 3090. Each image in the input video clip is first augmented by a randomly selected operation from a set of image transformations, including random resize crop, color jitter, grayscale, and Gaussian blur. The augmented video clip is then passed into the vision backbone, and the resulting features are flattened into 512-dimensional feature vectors. The transformer encoder, on the other hand, comprises 8 stacked layers of transformer encoder layers, with each layer employing 4 heads. The dimension of the feedforward network is set to 512.

The prototype layer $f_{\text{prototype}}$ is implemented as a single linear layer without bias, and we normalize its weights with every training iteration. We freeze its weights for the first 3 training iterations to stabilize the training process. For the TCN loss, in practice, we replace the skill prototype probability with its unnormalized version $z_t^{\text{T}} C$ (before applying the Softmax function). We noticed that the Softmax function saturates the gradient, leading to unstable training.

The additional hyperparameters are summarized in Table A5 and Table A6 for simulated and real-world kitchen environments, respectively.

Table A5: **Simulated Kitchen Skill Discovery Hyperparameter**

| Hyperparameter | Value |
|---|---|
| Video Clip length $l$ | 8 |
| Sampling Frames $T$ | 100 |
| Sinkhorn iterations | 3 |
| Sinkhorn epsilon | 0.03 |
| Prototype loss coef | 0.5 |
| Prototype loss temperature | 0.1 |
| TCN loss coef | 1 |
| TCN positive window $w_p$ | 4 |
| TCN negative window $w_n$ | 12 |
| TCN negative samples | 16 |
| TCN temperature $\tau_{\text{tcn}}$ | 0.1 |
| Batch Size | 16 |
| Training iteration | 100 |
| Learning rate | 1e-4 |
| Optimizer | ADAM |

### A.4.3 Skill Alignment Transformer

The Skill Alignment Transformer (SAT) comprises a state encoder, denoted as $f_{\text{state-encoder}}$, and a transformer encoder. The state encoder is implemented as standard Resnet18. The transformer encoder consists of 16 stacked layers of transformer encoder layers, each employing 4 heads. and the feedforward network has a dimension of 512. As depicted in Section 3.3 of the paper, a set of skill representations $\{z_t\}_{t=0}^{T^i}$ is extracted from the sample trajectory $\tau_i$ and passed into SAT as skill tokens. For practical purposes, XSkill adopts a uniform sampling approach, selecting $N_{\text{SAT}}$ prototypes from the skill list. This approach is motivated by two primary reasons. First, skills are typically executed over extended periods, and we only require information about the start and end times, as well as the time allocated to each skill. Uniform sampling preserves this necessary information while reducing redundant prototypes in the list. Second, human demonstrations may occur at a significantly faster pace than the robot's execution, leading to variations in the length of the skill list. This discrepancy can hinder the learning algorithm's performance during inference. By uniformly sampling a fixed number of frames from the set, the learning algorithm operates under consistent conditions in both learning and inference stages. $N_{SAT}$ is set to approximately half of the

Table A6: **Realworld Kitchen Skill Discovery Hyperparameter**

| Hyperparameter | Value |
|---|---|
| Video Clip length $l$ | 8 |
| Sampling Frames $T$ | 100 |
| Sinkhorn iterations | 3 |
| Sinkhorn epsilon | 0.03 |
| Prototype loss coef | 0.5 |
| Prototype Softmax temperature | 0.1 |
| TCN loss coef | 1 |
| TCN positive window $w_p$ | 6 |
| TCN negative window $w_n$ | 16 |
| TCN negative samples | 16 |
| TCN temperature $\tau_{\text{tcn}}$ | 0.1 |
| Batch Size | 20 |
| Training iteration | 500 |
| Learning rate | 1e-4 |
| Optimizer | ADAM |

average length of frames in robot demonstrations. During inference, if the length of the extracted skill list is less than $N_{SAT}$, XSkill uniformly up-samples the skill list. We include a representation token after the skill token and the state token to summarize the prediction information. The latent representation of the representation token is then passed into a multi-layer perceptron (MLP) to predict the desired skill $z$. We set $N_{\text{SAT}} = 100$ in the simulated kitchen environment and $N_{\text{SAT}} = 200$ as the realworld robot trajectories is significantly longer than those in simulation.

### A.4.4 Diffusion Policy

We use the original code base from Chi et al. [35] and adapt same the configuration for both the simulated and realworld environment. We refer the reader to the paper for details.

Listing 1: Pseudocode for Sinkhorn

```
"""
PyTorch−like pseudocode for Sinkhorn−Knopp
"""
# Sinkhorn−Knopp
def sinkhorn(scores, eps=0.05, niters=3):
    Q = exp(scores / eps).T
    Q /= sum(Q)
    K, B = Q.shape
    u, r, c = zeros(K), ones(K) / K, ones(B) / B
    for _ in range(niters):
        u = sum(Q, dim=1)
        Q *= (r / u).unsqueeze(1)
        Q *= (c / sum(Q, dim=0)).unsqueeze(0)
    return (Q / sum(Q, dim=0, keepdim=True)).T
```

## A.5 Environment&Data Collections

We begin with a formal description of the three distinct data sources. **1). Human demonstration dataset**: Herein, $\tau_i^h = \{o_0, .., o_{T_i}\}$, where $o_t$ denotes the RGB visual observation at the time $t$. Within each trajectory, a subset of skills $\{z_j\}_{j=0}^{J_i}$ is sampled from a skill distribution $p(\mathcal{Z})$ containing $N$ unique skills and a human performs in a random sequence. **2). Robot teleoperation data**: This dataset comprises teleoperated robot trajectories $\tau_i^r = \{(o_0, s_0^{prop}, a_0), .., (o_{T_i}, s_{T_i}^{prop}, a_{T_i})\}$, where $s_t^{prop}, a_t$ correspond to robot proprioception data and end-effector action at time $t$ respectively. We utilize $s_t$ as the symbol for $o_t, s_t^{prop}$ throughout the main paper. Analogous to the human

demonstration dataset, each trajectory incorporates a subset of skills $z_j{}_{j=0}^{J_i}$, sampled from the skill distribution $p(\mathcal{Z})$, which the robot executes in a random sequence. **3). Human prompt video**: This single trajectory of human video $\tau_{prompt}^h = \{o_0, .., o_{T_{prompt}}\}$ demonstrate **unseen** composition of skills $\{z_j\}_{j=0}^{Jprompt}$ taken from the skill distribution $p(\mathcal{Z})$. We represent the RGB video trajectories that include only the RGB visual observation $\{o_0, .., o_{T_i}\}$ for both human and robot in the main paper as $V_i$ for the sake of simplicity.

## A.6 Simulation

In order to produce a cross-embodiment dataset, we modified the initial Franka Kitchen setup, introducing a sphere agent distinctly visual from the original Franka robot. The sphere agent demonstration dataset was generated by substituting the Franka robot arm with the sphere agent and re-rendering all 600 Franka demonstrations. The images for both Franka robot and sphere agent demonstration are in a resolution of 384x384. Each trajectory in this dataset features the robot completing four sub-tasks in a randomized sequence. The demonstration from both embodiments was divided into a training set and a prompt set, with the latter containing trajectories involving unseen combinations of sub-tasks. This requires the robot to complete tasks namely, opening the microwave, moving the kettle, switching the light, and sliding the cabinet in order.

For skill discovery, we downsampled the demonstration videos to a resolution of 112x112 and randomly applied color jitter, random cropping, Gaussian blur, and grayscale to the input video clips. For diffusion policy training, the environment observation incorporates a 112 x 112 RGB image and a 9-dimensional joint position (include gripper). We used a stack of two consecutive steps of the observation as input for the policy.

## A.7 Realworld

We conducted data collection for our cross-embodiment dataset in a real-world kitchen environment using a UR5 robot station. The UR5 robot is equipped with a WSG50 gripper and a 3D printed soft finger. It operates by accepting end-effector space position commands at a rate of 125Hz. In the robot station, we have installed two Realsense D415 cameras that capture 720p RGB videos at 30 frames per second. One camera is mounted on the wrist, while the other provides a side view.

Our dataset consists of demonstrations involving both human and robot teleoperation for four specific sub-tasks: opening the oven, grasping cloth, closing the drawer, and turning on the light. To introduce variability, the initial locations of the oven and the pose of the cloth are different for each trajectory. Each demonstration trajectory involves the completion of three sub-tasks in a random order.

For training, we created seven distinct tasks for each embodiment and collected 25 trajectories for each task. The robot teleoperation demonstrations were recorded using a 3Dconnexion SpaceMouse at a rate of 10Hz. For the inference task, we created two unseen tasks with three sub-tasks each, and four unseen tasks with four sub-tasks each. For tasks with three sub-tasks, we recorded both human and robot demonstrations as prompt videos, while for tasks with four sub-tasks, we recorded human demonstrations only. The details of task collections are illustrated in Tab. A7

During skill discovery, we exclusively utilized videos recorded from the side camera and downsampled them to 160x120 at 10fps. Similar to before, we applied random transformations such as random crop, Gaussian blur, and grayscale to the input video clips. For diffusion policy training, we used visual inputs from both cameras, downscaled to 320x240. The input to the diffusion policy included a 6-dimensional end effector pose, a 1-dimensional gripper width, and two visual inputs from both cameras. We only considered one step of observation as the policy input. Position control was selected as the diffusion-policy action space, encompassing the 6-dimensional end effector pose and 1-dimensional gripper width. During training, we applied random crop with a shape of 260x288, while during inference, we utilized a center crop with the same shape.

Table A7: **Training & Inference Task**

| | Tasks | Human(seconds) | Robot(seconds) |
|---|---|---|---|
| Overlapping Training Task | Draw, Light, Oven | 12.92 + 1.19 | 29.12 + 2.03 |
| | Light, Cloth, Oven | 15.23 + 0.92 | 32.76 + 2.42 |
| | Draw, Light, Cloth | 15.73 + 1.22 | 26.83 + 2.28 |
| | Draw, Cloth, Light | 17.21+1.05 | 31.71 + 3.74 |
| Human exclusive Training Task | Oven, Draw, Cloth | 11.62+1.58 | / |
| | Cloth, Oven, Light | 12.69+0.86 | / |
| | Cloth, Light, Oven | 13.37+0.67 | / |
| Robot exclusive Training Task | Light, Oven, Draw | / | 32.41+3.04 |
| | Oven, Light, Cloth | / | 26.75+2.56 |
| | Light, Draw, Cloth | / | 27.10+1.95 |
| Inference Task | Oven, Draw, Cloth | 14.4 | 45.7 |
| | Draw, Cloth, Oven | 12.6 | 41.4 |
| | Oven, light, Cloth, Draw | 20.5 | / |
| | Draw, Cloth, Light, Oven | 20.9 | / |
| | Draw, Oven, Cloth, Light | 21.0 | / |
| | Draw, Light, Cloth, Oven | 19.2 | / |

