# OpenReview forum: "XSkill: Cross Embodiment Skill Discovery"
_robot-learning.org/CoRL/2023/Conference — CoRL 2023 Poster_

### Official Review · Reviewer_9Eu4 · 2023-07-08

**Confidence:** 4
**Originality:** Good
**Technical Quality:** Good
**Clarity Of Presentation:** Fair
**Impact:** 3

**Recommendation:**

Weak Accept: I recommend accepting the paper, but will not argue for my recommendation if the majority of other reviewers have a different opinion.

**Review:**

++: Strength, --: Weakness

**Novelty & Significance**

++ Although bearing similarities with previous works on latent space encoding for cross-domain imitation learning, XSkill is interesting as it discovers and recognizes the primitive skills across morphologies, which is a powerful concept for cross-domain imitation learning for long-horizon tasks.

++ Skill Alignment Transformer is a good idea for re-planning on task-level.

++ The real-world kitchen dataset will be appreciated by the community if released.

-- It seems XSkill still needs datasets collected from teleoperations. Similar to telekinesis teaching, these datasets are known to be very hard to collect large samples. This will hinder the applicability of XSkill.

-- Other limitations are mentioned by the authors.

**Quality & Clarity**

The presentation flow of the paper is fair to follow. However, there are some confusing technical.

++ Strong experiment setups and adequate baselines.

-- It is very confusing on how the human & robot videos representing the same task are sampled and constrained to be closed in the latent space. Fig. 1 depicts aligned videos are input for learning latent space with the same task and different embodiments. However, on line 95, both human and robot datasets are said to be unaligned.

-- Fig 2. does not help in the understanding of training XSkill. What is “augutation”? Why are there two versions of video batch for computing the skill probability vectors?

-- It is not clear how the Sinkhorn Clustering works in this case. On line 144, what are C and Z, as it is not introduced? Are they input as costs or marginal probabilities? It is very confusing on the intuition at lines 146-148. Why does utilizing as many as possible skill prototypes promote prototype alignments in different embodiments? Overall, there is also no clear motivation why Sinkhorn Clustering is applied in this case.

-- The motivation for using a low-level diffusion policy in this setting is unclear. Why does multi-modality in myopic policy benefit in executing tasks? It seems myopic diffusion policies might jump between modes and thereby deteriorate task success.

-- Metrics on overall task success are needed, as this is what really matters.

-- Some typos exist. For example, “CorssEntorpy” on line 134 in Fig. 2 caption.

**Quality Of The Limitations Section:**

Additional details required

**Questions For Rebuttal:**

Overall, I encourage the authors to significantly refine the papers, as it is very unclear in their current form.

1. An overall pseudo code for training and inference of XSkill would be appreciated.
2. More elaborations on the Sinkhorn Clustering motivation and how it is used to compute target skill probability from batch videos are needed.
3. What are the motivations for using a low-level diffusion policy in this case?
4. With the described training process, it seems XSkill will overfit the objects being manipulated in the videos, hence bypassing the correspondence problem between morphologies. Is this really a desirable property? If so, in the imitation learning case, we can just learn the affordance of objects being manipulated in keyframes and no need for video data. Otherwise, evidence for capturing morphology correspondence must be provided, e.g., a rotating human hand with a corresponding rotating robot end-effector.

**Robotics Focus:**

Sufficient demonstration on hardware

**Summary Of Paper:**

This paper proposes XSkill - an imitation learning method for cross-embodiment settings. XSkill recognizes the skill sequences from a human demonstration video as a prompt, then conditions the skill sequences on a low-level diffusion policy to reproduce the task. At training phase, XSkill samples two aligned human and robot videos from a dataset, maps them into a latent space, predicts the primitive skill probability vectors from the observed latent vector, then finally matches the skill probability vectors with Sinkhorn Clustering targets by a CrossEntropy objective. Additionally, a low-level diffusion policy is trained with the conditioned skill latent representation to execute the low-level skills. At inference phase, a human video is prompted together with the current robot observation, then XSkill predicts the skill sequences and the current skill to condition on the low-level diffusion policy for task execution. The method's efficacy is shown in various real-world kitchen tasks.

**Summary Of Recommendation:**

Overall, the method has some originality and good experiments showing its efficacy. However, with the current form of paper quality and presentation, I recommend rejection.

------ Post Rebuttal ------

The authors addressed all my concerns. I decided to raise the score to Weak Accept.

---

### Official Review · Reviewer_fc5Y · 2023-07-15

**Confidence:** 4
**Originality:** Very Good
**Technical Quality:** Very Good
**Clarity Of Presentation:** Good
**Impact:** 4

**Recommendation:**

Strong Accept: I recommend accepting the paper and will argue for my recommendation even if other reviewers hold a different opinion.

**Review:**

This work shows impressive performance at imitating human video by extracting and composing cross-embodiment skills. The method and  results are clearly presented. The authors discuss why each component of the method is needed and how they build on past work.

Strengths:
- Results show good performance at one-shot imitation of novel combinations of skills from human video.
- Videos show good performance at imitation and replanning with perturbations
- Approach is robust to changes in execution speed across embodiments due to SAT component
- Writing is clear and video on website explains method well
- Figure 5 shows the structure of extracted skill decompositions well
- The approach extracts skill decompositions and aligns them across robot and human video without supervision
- Policy and skill learning only needs offline data

Weaknesses:
- Number of skills must be specified in advance
- Limited number of concrete skills in the evaluation environments: 7 in simulation and 4 in the real world
- There are many components in the pipeline for this approach, not all of which are ablated / justified (see below)

**Quality Of The Limitations Section:**

Limitations are addressed clearly

**Questions For Rebuttal:**

- Why is the time contrastive loss needed for the approach / do any experiments show it is helpful?
- The prediction of skills for each frame could be clarified with pseudocode or a diagram. The learned policy appears to be conditioned on predicted skill rather than the nearest prototype, which seems more natural if the skill predictions are justified as classifying between prototypes. Was this choice ablated?
- What does the variation between prototypes within the same cluster of Figure 5 represent?


**Robotics Focus:**

Sufficient demonstration on hardware

**Summary Of Paper:**

The paper proposes an approach for imitating human demonstrations by using cross-embodiment skill representations. The approach learns skill representations from segments of video clustered through self-supervised augmentations. Alignment across human and robot skills is encouraged by maximizing the entropy of the marginal skill distributions predicted within each embodiment. To imitate human videos, a transformer is trained to predict a next skill from the human video to perform at a given state. A skill-conditioned policy is learned to execute these skills sequentially. Experimental results in simulation and a real-world setting show the robot can imitate novel combinations of skills given with human video. The pipeline outperforms ablations and is robust to perturbations and changes in execution speed across embodiments.

**Summary Of Recommendation:**

The paper presents a complex but well-justified approach to imitating human videos through hierarchical skill decomposition. Unlike past work, the approach is able to discover skills that transfer across embodiments without supervision or explicit alignment. Experiments ablate the components of the method and show reliable performance in simulation and the real world with robustness to speed and re-planning on perturbation. Limitations include using a small set of skills in both settings and some minor issues with the clarity and justification of the approach.

---

### Official Review · Reviewer_1ysS · 2023-07-20

**Confidence:** 3
**Originality:** Good
**Technical Quality:** Very Good
**Clarity Of Presentation:** Good
**Impact:** 3

**Recommendation:**

Weak Accept: I recommend accepting the paper, but will not argue for my recommendation if the majority of other reviewers have a different opinion.

**Review:**

The technical components of the paper seems solid and well executed. The authors put substantial effort in building such a complex system, and evaluate the system on both synthesized and real tasks.

In terms of the core idea of “skill discovery”, I’m not too sure if calling them “skills” is a bit of over-claim, and in the end if they are better (more generalizable, robust) than learning a shared latent continuous encoding of human and robot videos. In detail:
1. There are 512 skill prototypes and the paper is “evenly assigning” samples in each batch to all prototypes — most of the skills learned therefore seem to only represent short segments in the video without interpretable meaning.
2. If most of these 512 skills are not interpretable anyways, one might wonder if one should build an easier system with continuous (rather than discrete as in the paper) latent representations. There is plenty of literature learning correspondence between images/video clips between two domains, which can potentially map human and robot video windows to a shared latent space. The policy then might directly operate on current latent (mapped from robot video window) and near-future latent from human video. While asking a new baseline is out of scope, I might appreciate some discussions in the paper on system design alternatives.

Other minor concerns:
1. The discussion of related works is not entirely fair. To my understanding, though it’s true that MimicPlay [61] does not take human prompt video as input, the focus is also quite different with [61] trying to minimize robot demo collection which is much more tedious than human demos. This paper seems to use same amount of human and robot demo data, which I assume means more robot demo time than [61]
2. The paper honestly describes that when the human prompt video contains unseen transitions between temporal tasks, the system would fail. Then what exactly is the generalization ability of this learned representation? I don’t feel like we can confidently call a task sequence “unseen” if both the primitive tasks (e.g. press button) and the transitions between tasks need to be “seen”.
3. Even combing the main text and appendix, the paper is not self-contained in some places. Quick examples: what are the augmentations used? What is “iteratively modified to produce a double stochastic matrix” and why this can be intuitively viewed as “entropy regularization”?

**Quality Of The Limitations Section:**

Limitations are addressed clearly

**Questions For Rebuttal:**

See above.

**Robotics Focus:**

Sufficient demonstration on hardware

**Summary Of Paper:**

The paper proposes a new system for imitation learning from human videos. The input during training time is the demonstration video of both the human and the robot performing same set of tasks (with temporal compositions), and during run time the system takes one single video of human demonstration (possibly with unseen temporal composition) to specify task sequence order, and a continuous stream of robot observation for controller feedback (using a Diffusion policy). The core innovation of the paper, is to learn the correspondence between robot and human demo video by mapping them to a same set of “skill prototypes”. The network loss encourages a batch of human or robot video clips to map to as many prototypes as possible, indirectly encouraging clips from human and robot containing similar temporal (i.e. task) information to map to the same prototype.

**Summary Of Recommendation:**

I lean towards acceptance given the effort the authors put into building such a system, and running thorough evaluations for it. On the other hand, I hold reservations on what benefit are we actually gaining, in terms of ability to generalize etc., with such large number of total learned skill prototypes.

---

### Author Response · Authors · 2023-08-10
**General Response**

We extend our gratitude to the reviewers for their insightful feedback. Their recognition of the novelty (R1, R2, R3) and technical solidness (R1, R2, R3) and impressiveness of our experiments (R2), is truly encouraging. Our revision contains the following major updates:

1. We have significantly improved the paper presentation by

   - Added the motivation to use Sinkhorn clustering during the skill discovery phase (Section 3.1) and its technical details (appendix A4.1).
   - Added the pseudo-code for the training and inference stages for XSkill (appendix A4).
   - Fixed unclear information in Fig. 1 and Fig. 2.
   - Added the overall task success rate as an additional metric in experiments.
   - Included more training details, e.g., augmentation operation.

2. Additional experiments to validate XSkill’s generalization ability to unseen transitions (R1) and applicability (R3). Our experiments (appendix A1) suggest that XSkill can generalize to new tasks involving both previously unseen skill compositions and unseen transitions. This further provides evidence for XSkill’s applicability, as our framework can one-shot generalize to unseen tasks with only human prompt videos.

3. Additional experiments to validate the benefits of contrastive loss (R2). Our experiments (Section 4) suggest that time contrastive loss is crucial for extracting skills and capturing the temporal effects of skills.

We have highlighted our major updates in blue and attached the appendix to the main paper.

We address the reviewer's individual comments below.

---

### Decision · Program_Chairs · 2023-08-30

**Decision:**

Accept (Poster)

**Comment:**

This work introduced an imitation learning algorithm that can perform tasks based on a single human demonstration (prompt video). The key technique is constructing a latent skill representation space (XSkill space) shared across human and robot embodiments. This skill space is learned through self-supervised objectives, and the skill representation is used to condition a low-level diffusion policy to generate the motor actions. This paper received mixed initial reviews. The reviewers raised some concerns, including the presentation clarity, comparisons to existing work like MimicPlay, the use of discrete skill prototypes, etc. The authors did a good job addressing these issues in the discussion phase. As a result, Reviewer 9Eu4 updates their rating to Weak Accept, making the final recommendations from the reviewers all positive: two Weak Accepts and one Strong Accept. Although some of the concepts behind this work have been explored in prior work, the AC believes that this work has presented a convincing system with the ability to understand and execute tasks based on human prompt videos. Taking into account the reviews, the rebuttal, and post-rebuttal discussions, the AC recommends accepting this paper at CoRL.